# HeProMo: A decision support tool to estimate wood harvesting productivities

**Stefan Holm** *, **Fritz Frutig, Renato Lemm, Oliver Thees, Janine Schweier**

Swiss Federal Institute for Forest, Snow and Landscape Research WSL, Birmensdorf, Switzerland

* stefan.holm@wsl.ch

## Abstract

In the field of forestry, one of the most economically important ecosystem service is the provision of timber. The need to calculate the economic effects of forest management in the short, medium, and long term is increasing. Forest operations or timber harvesting, which comprises felling, processing, and transport of trees or timber, are responsible for a large part of the costs and environmental impacts associated to forest management or enterprises. From a decision maker's perspective, it is essential to estimate working productivity and production costs under given operating conditions before any operation is conducted. This work addresses the lack of a valid collection of models that allows estimating time, productivities, and costs of labor and machinery for the most important forest operations in forest stands under Central European conditions. To create such models, we used data from forest enterprises, manual time studies, and the literature. This work presents a decision support tool that estimates the wood harvesting productivities of 12 different kinds of forest operations under Central European conditions. It includes forest operations using chainsaws, harvesters, skidders, forwarders, chippers, cable and tower yarders, and helicopters. In addition, the tool covers three models for wood volume estimation. The tool is written in Java and available open-source under the Apache License. This work shows how the tool can be used by describing its graphical user interface (GUI) and its application programming interface (API) that facilitates bulk processing of scientific data. Carefully selected default values allow estimations without knowing all input variables in detail. Each model is accompanied by an in-depth documentation where the forest operation, input variables, formulas, and statistical background are given. We conclude that HeProMo is a very useful tool for applications in forest practice, research, and teaching.

## 1. Introduction

Sustainable forest management (SFM) has become one of the core concepts in the use and conservation of forest resources, thus in managing forest ecosystems and their services [1]. In general, Ecosystem Services (ES) are the products of functioning ecosystems that benefit people [2], and in the field of forestry, one of the most economically important ES is the provision of timber (e.g., round wood, pulp and paper, energy wood). Other ES have regulating (e.g.,

**Data Availability Statement:** The software (executable jar-file) can be downloaded at: https://github.com/holmstefan/hepromo/releases The source code can be accessed at: https://github.com/holmstefan/hepromo.

**Funding:** The present work was partially funded by the Swiss Federal Institute for Forest, Snow and Landscape Research (WSL) and the Swiss Federal Office for the Environment (FOEN). The funders had no role in study design, data collection and analysis, decision to publish, or preparation of the manuscript.

**Competing interests:** The authors have declared that no competing interests exist.

climate stabilization), cultural (e.g., recreation), and supporting (e.g., nutrient cycling) functions [3]. The provision of wood products and other ES depends on various aspects. Because the majority of ES are public goods and they lack of markets, their provision strongly depends on forest ownership structures and political frameworks [4, 5].

In this context, the requirements of the society on the forest and the complexity of decision-making in forestry increase. Therefore, the need to calculate the economic effects of forest management in the short, medium, and long term is increasing. Further, forests management strategies significantly influence the amount and quality of provided wood and other ES. When silvicultural prescriptions are implemented, different Forest Operations (FOs) might be more or less favorable for a decision maker. FOs or timber harvesting, which comprises felling, processing, and transport of trees or timber, are responsible for a large part of the costs (approximately 40–60% of the total expenses of a forest enterprise in Switzerland) and environmental impacts associated to forest management or enterprises. Three common and important harvesting methods in Central Europe include full-tree (FT), tree-length (TL), and cut-to-length (CTL) [6, 7]. When applying the FT method, trees are felled and transported to roadsides with branches and top intact. The TL method requires felling, delimbing (removing branches), and topping (cutting the top of the stem at a specified diameter) before skidding to a truck-accessible road. The CTL method involves the same processing as TL, but with addition of stem bucking (cutting a stem into logs of different specified lengths) before hauling from the stump to a truck-accessible road. All methods can be deployed with different levels of mechanization: motor-manually (MM), semi-mechanized (SM), or fully mechanized (FM).

From a decision maker's perspective, it is essential to estimate working productivity and production costs under given operating conditions before any operation is conducted. This is needed from operating and strategic viewpoints. Therefore, the working time and costs of FOs as well as other aspects such as environmental impacts have to be estimated for entire enterprises or larger regions and an extended period of time.

There are various models to calculate the costs of *individual* harvesting methods and processes (e.g., [8, 9]). Some models can also be used to estimate the costs of full supply chains (e.g., Holzernte 8.0 by [10]). However, these models require productivity to be provided as input data. Further, the models are valid only for a specific region and need to be customized when applied on an enterprise level (e.g., [11]). Therefore, this work addresses the lack of a valid collection of models that allows estimating time, productivities, and costs of labor and machinery for the most important FOs in forest stands under Central European conditions. Such a collection could be used as a reference in research, teaching, and practice.

Consequently, our approach was to develop such a tool. In this paper, we aim to

1. present the open-source software HeProMo and its development from an operational and software engineering perspective;

2. demonstrate how it can be used in business and academics for either estimating an individual FO or bulk processing large FO data sets;

3. derive future possibilities for the use and development of the software.

The software, which was developed within the last 20 years [12], is a collection of models each representing a certain FO.

## 2. Methods

HeProMo models represent FOs typically conducted in Central European forests. The forests in these regions are characterized by rather small plots of land, high share of public owned

forests, multifunctional silviculture management as well as mixed and uneven-aged stands. The FOs in these regions are characterized by single tree thinning and comparable small timber volumes per cut, thus resulting in frequent machinery transport and fewer productive machine hours. Further, many restrictions such as soil compaction and nutrient extraction need to be considered. Each HeProMo model represents a processing or a transport operation —ground-, cable- or air based—and is accompanied by an in-depth pdf-documentation where the FO, input variables, formulas, and statistical background are described. The main output variable calculated by each model is productivity, measured in cubic meters of wood processed per hour. Based on productivity, time and costs of labor and time and costs of machines are calculated. Costs are displayed as costs per cubic meter and total costs of FO.

The following subsections present an overview of the modeled FOs (section 2.1), data collection process (section 2.2), model building process (section 2.3), and information about the concepts and characteristics of the data units used in HeProMo (section 2.4).

## 2.1 Modeled forest operations and volume estimations

The modeled FOs can be divided in processing models, transportation models, and mixed models (Table 1). In addition to these models, three volume estimators are included in the tool:

- **Biomass estimator**: estimates the volumes of stem wood, compact branch wood, brushwood, and mass of needles and leaves for a set of trees based on tree species and tree location characteristics.

- **Energy wood estimator**: estimates the volumes of energy wood from different parts of a tree, based on the characteristics of a forest area (e.g., proportion of broadleaves, mean diameter at breast height (dbh), and FO (e.g., percentage of logging waste).

- **Stem wood estimator**: estimates the stem volume of a single tree based on tree and location characteristics.

**Table 1. Modeled forest operations.**

|  | Model name | Modeled forest operation |
|---|---|---|
| **Processing Models** | Motor-manual felling and processing | Felling, debranching, measuring, and cutting into logs, using a chainsaw. |
|  | Harvester | Felling, processing of trees into logs (debranching, measuring, cutting), and placing branches on the skid trail. |
|  | Chipper | Chipping of crowns and branches, located at a truck-accessible road, into a container. |
| **Mixed Models** | Wood chips transport | Transport of wood chips to the customer. Wood chips are either already loaded in a container (lifting system) or chipped into a container mounted on a transport vehicle. |
|  | Felling and pre-skidding | Felling using a chainsaw and pre-skidding of full trees to the skid trail. |
| **Transportation Models** | Skidder | Skidding of felled logs from the stand to a storage place at a truck-accessible road. |
|  | Forwarder (roundwood only) | Hauling of logs lying at the skid trail (CTL) to a storage place at a truck-accessible road. |
|  | Forwarder (roundwood and energy wood) | Hauling of logs and energy wood assortments to a storage place at a truck-accessible road. Productivity, time, and costs are differentiated by roundwood and energy wood. |
|  | Long distance cable yarder | Mounting of yarder, yarding of logs from the stand to the forest road using a long distance cable yarder, and dismounting the yarder. |
|  | Tower yarder | Mounting of yarder, yarding of logs from the stand to the forest road using a tower yarder, and dismounting the yarder. |
|  | Tower yarder with mounted processor | Felling by chainsaw, yarding of full trees by tower yarder to the forest road where they are debranched and cut by a mounted processor, and moving logs to a storage place by a separate engine. |
|  | Helicopter | Transportation of logs or full trees from the stand to a truck-accessible road by helicopter. |

## 2.2 Data collection

The first digitized productivity models of HeProMo in the early 2000s were derived from existing calculation tools in paper form, such as long distance cable yarder [13]. Their data base was founded on manual time studies. Manual time studies can provide detailed information for an individual case to be recorded. However, they require a corresponding expenditure of time by the personnel who conducted the time studies, in particular where a statistical analysis of data is foreseen. Depending on the demand for representativeness of results, they are associated with high time requirements and costs. In practice, this usually leads to a small amount of data. To overcome the time, quality, and financial constraints, we therefore had to find other methods to obtain data of harvesting processes.

To obtain the data required to update our productivity models in a timely manner with sufficient quality, we used operational data. Hence, most productivity models updated from 2014 are based on data from forest enterprises in Germany and Switzerland (Table 2). Two models, chipper and wood chips transport, were built with data from tests published in forest literature [14, 15]. The productivity models for long distance cable yarder, tower yarder, and helicopter have not yet been updated. Despite being based on rather aged data, there is no urgent need to update these models, as they still fit well.

The procurement of performance data from forest enterprises or service providers is not easy due to competition, obligation of secrecy, and data protection, but also simply due to the lack or restriction of accounting. In general, it is easier to obtain data from public than from private companies. If data are available, they are often large datasets (especially compared to

**Table 2. Data sources for different productivity models in HeProMo version 2.4.**

| Forest operation | Data origin | Data range | Reference and country of reference | Year of actualization |
|---|---|---|---|---|
| Motor-manual felling and processing | Forest enterprises | 21'879 cuts, 8'000'000 m$^3$ | Bavarian State Forestry Department, Germany | 2014 / 2017 |
| Harvester | Forest enterprises | 928 cuts, 9 harvester types, 533'639 m$^3$ | State-owned forestry technical bases (Rhineland-Palatinate and Baden-Wuerttemberg), Germany | 2014 |
| Chipper | Publications | 5 chipper types, 38 cuts, 4 chipping methods | [14], Germany | 2018 |
| Wood chips transport | Publications and small survey | 3 transport companies | [15], Switzerland | 2018 |
| Felling and pre-skidding | Forest enterprises | Tariff d1, combined method with pre-skidding and processing of full trees | State-owned forestry technical base (Baden-Wuerttemberg), Germany | 2018 |
| Skidder | Forest enterprises | 277 cuts, 5 skidder types, 90'565 m$^3$ | State-owned forestry technical bases (Rhineland-Palatinate and Baden-Wuerttemberg), Germany | 2014 |
| Forwarder (roundwood only) | Forest enterprises | 336 cuts, 14 machine types, 450'000 m$^3$ | State-owned forestry technical bases (Rhineland-Palatinate and Baden-Wuerttemberg), Germany | 2017/18 |
| Forwarder (roundwood and energy wood) | Forest enterprises / publications | Roundwood: 450'000 m$^3$ <br> Energy wood: meta-analysis | Roundwood: State-owned forestry technical bases (Rhineland-Palatinate, Baden-Wuerttemberg, Thuringia), Germany <br> Energy wood: [16], Germany | 2017/18 |
| Long distance cable yarder | Time studies Existing model: [13] | 44 cuts | 44 forest enterprises and private forest operators, Switzerland | 1999 |
| Tower yarder | Time studies | 22 cable lines, 699 load cycles, ~1800 m$^3$ | 22 forest enterprises and private forest operators, Switzerland | 1999 |
| Tower yarder with mounted processor | Forest enterprises | 104 cuts, 2 machine types, 66'762 m$^3$ | Forest enterprise, Switzerland | 2017/18 |
| Helicopter | Publications | 3 helicopter classes | [17, 18], Switzerland | 2003 |

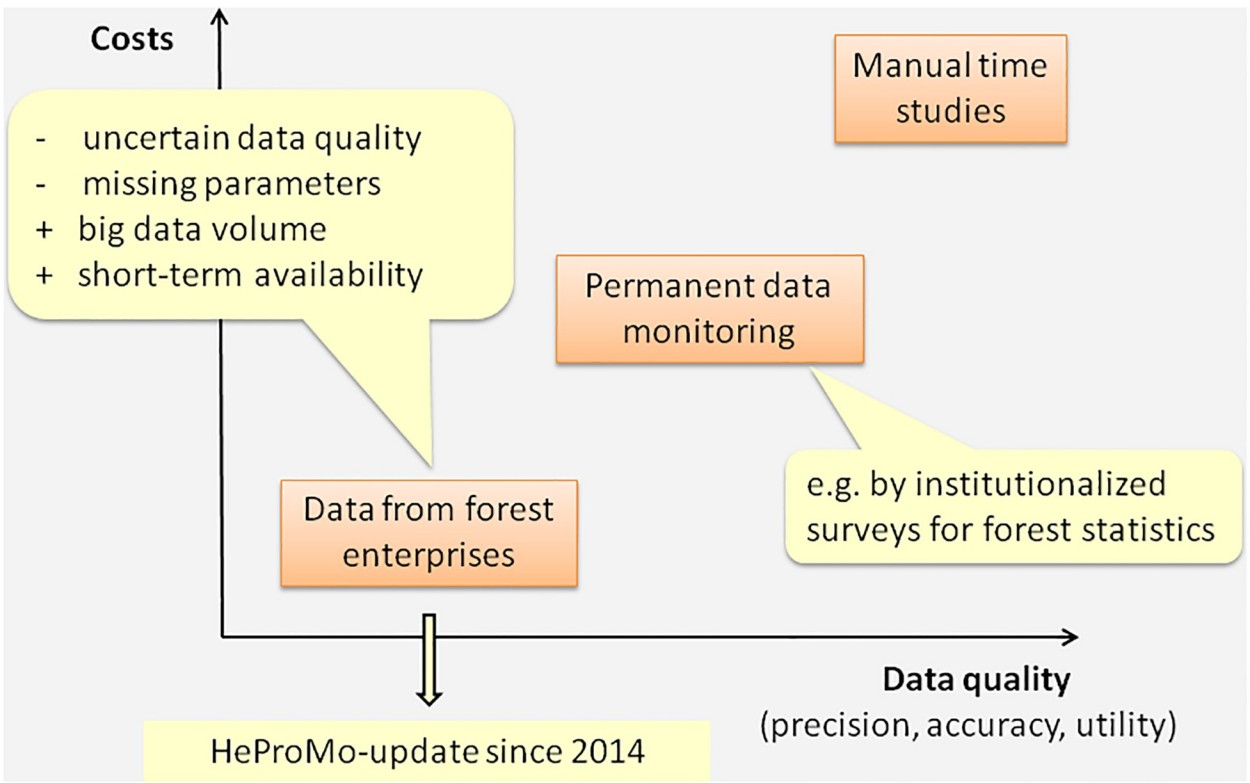

**Fig 1. Relation between data quality and costs for different data collection methods (based on Thees & Frutig [19]).**

time studies), which allows their statistical analysis. However, the quality of these data cannot be assessed, or only to a limited extent, and performance-determining parameters may be missing from such datasets because the data were collected for other purposes.

Some required parameters may be missing in data sets from forest enterprises because these data have been gathered for uses other than building productivity models (Fig 1). How these cases were handled is described in the next section. Even with some parameters missing, well-fitted productivity models can be produced due to the large data volume (timber volume and time consumption covered by the data).

## 2.3 Modeling

The development of each HeProMo model follows a uniform construction plan with a three-shell structure (Fig 2). This three-shell structure is important for the flexibility of the use of the models regarding their spatial and temporal context and their maintenance and updating. The core is, in most cases, a statistical model, i.e., a regression model of productivity, derived from the available data with the help of the open-source statistical analysis software R [20] (a few models, chipper, woodchips transport, helicopter, and the volume estimators, are based on information from the literature and expert knowledge, not on an own evaluation of a larger statistical data set). The modeling results are documented and can be tracked by the user. For example, for the skidder model, the influences of average volume, machine type, and logging distance were statistically significant. In the second shell, the selected parameters are known to be relevant for productivity but have not been represented in the given data. Therefore, in this case, the lateral dragging distance, which is another relevant parameter, was implemented as a

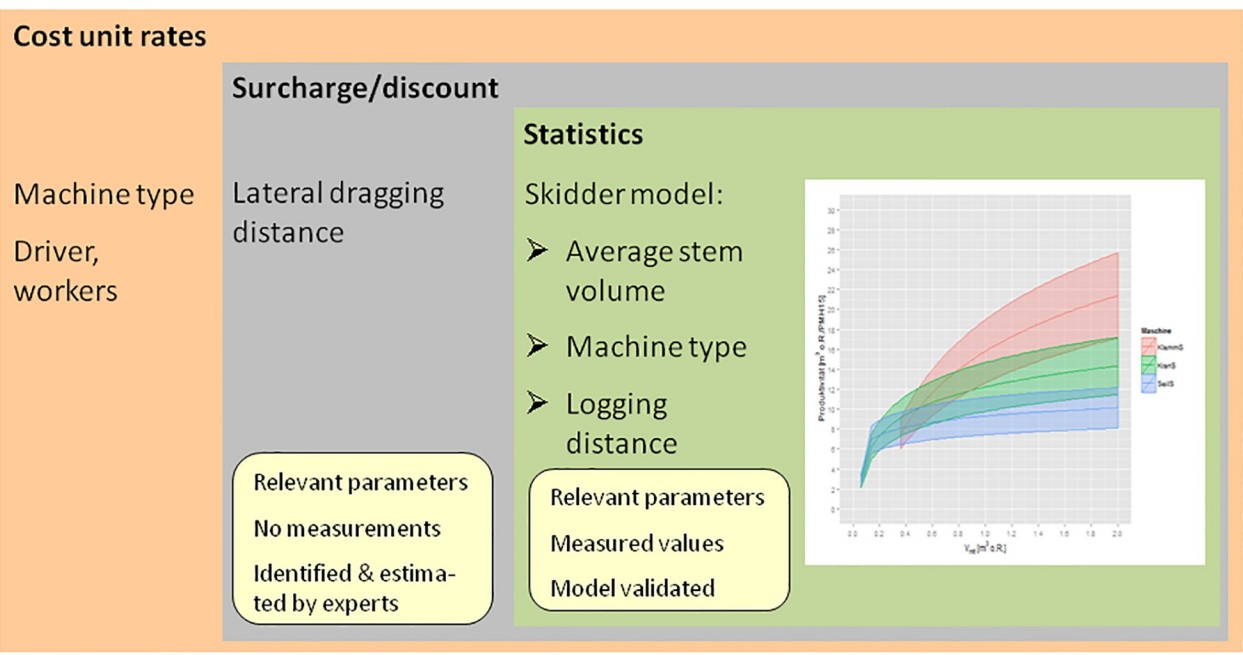

**Fig 2. Three-shell structure of the skidder model (based on Thees & Frutig [19]).**

discount that was estimated by experts from the forest practice. In other cases, we searched for information on influencing factors in the literature or in pre-calculation sheets established by administrations such as German state forest administrations (e.g., influence of skidding or forwarding distances on productivity [11]). In the third shell, the costs of production factors are introduced into the model.

## 2.4 Data units: Concepts and characteristics

**2.4.1 Time and productivity.** The time system used in HeProMo is based on Björheden & Thompson [21], modified by Heinimann [22]. The most relevant time units for HeProMo are the following, which are usually used in the result section of the models (Fig 3):

- *Workplace system hours* (**WPSH**): the work duration, i.e., the time between when the first person starts to work and the last person finishes working.

- *Workplace personnel hours* (**WPPH**): the total time the personnel needs to complete a task ("work time volume"). WPPH may be longer than WPSH if more than one person is working, as WPPH represents the sum of working time of all people working.

- *Productive machine hours* (**PMH15**): the amount of time that a certain machine was running, including interruptions up to 15 minutes (e.g., short maintenance times).

**2.4.2 Costs.** The cost unit rates for labor and machinery can be manually adapted to local conditions. The default monetary unit for the cost unit rates in HeProMo is Swiss Francs, abbreviated as CHF. The monetary unit can be set to an arbitrary value.

**2.4.3 Wood volumes.** The most important input variable for all models is the volume of wood to be processed. In most models, this is the merchantable wood volume, reported in cubic meters *over bark*. However, on the output side, the costs per cubic meter are usually

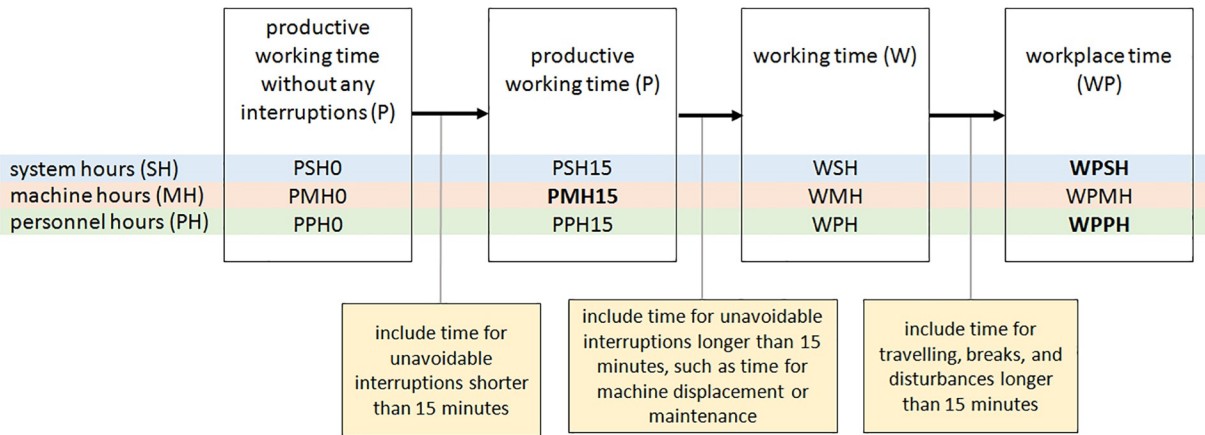

**Fig 3. Units used to describe the time of forest operations and their relations among each other.** Times can be calculated for the overall system, machinery, or personnel. The most used time units in HeProMo are in bold.

reported *under bark*. This reflects the fact that roundwood is measured over bark but is sold at forest roadside and customers pay the effective under bark volume only. The productivity is reported in both units, under and over bark, with the exception of energy wood related models, which are only reported over bark. The default value for the conversion factor from over to under bark is 0.9 (mean value of the most common tree species in Switzerland based on the Swiss National Forest Inventory—NFI) [23]. The conversion factor can be adapted in the user interface.

## 3. Software

HeProMo is a Java desktop application that can be executed without installation, the only requirement is to have a Java Runtime Environment (JRE) installed on the system. Currently, in 2020, the minimum required JRE version is 1.7. All models can be accessed with a graphical user interface (GUI) and via an application programming interface (API). The source code is available under the Apache License. The models have a varying number of input variables. To facilitate the use of the models, all input variables have reasonable, carefully selected default values. Thus, realistic calculations can be conducted even if some input variables are unknown to the user. Calculations are conducted "on-the-fly", which means that whenever an input variable is changed, the updated results are immediately shown to the user.

In the following subsections, after a short overview of the history of HeProMo (section 3.1), the access via GUI (section 3.2) and API (section 3.3) are explained, followed by technical information about the software architecture and testing procedure (section 3.4).

### 3.1 Development history

In 2000, eight productivity models were developed as individual COM (Component Object Model) software components [24, 25], which were incorporated into Excel spreadsheets. In 2003, a.NET application that contained all models within a single piece of software was developed. At the beginning of the 2010s, we decided for an overall update due to the appearance of new harvesting methods and the change in the productivity of FOs since the turn of the millennium. This led to a reimplementation of HeProMo as a Java application, which was made available in 2014. Since then, this version has been continuously maintained and further developed. By 2016, the software was available in English, German, French, and Italian. By 2019,

several new models were integrated into the software. These models cover additional FOs or replace already existing models because a newer data base was available. Finally, in 2020, HeProMo was converted into an open-source project, and an API has been made available in addition to the GUI. In the following subsections, we present the 2020 version in detail.

## 3.2 Graphical user interface

**3.2.1 Main window.** In the main window of HeProMo, the models are divided into three categories, namely i) Up-to-date models, ii) Models with out-dated data base, and iii) Volume calculation (Fig 4). The models with out-dated data base are available in German only, as they are no longer maintained but still provided to interested users (they have a higher number of input variables than the new models and therefore may be useful to investigate the influence of certain variables on productivity, even though the level of productivity or cost is outdated). The other two categories are described in detail in the following sections.

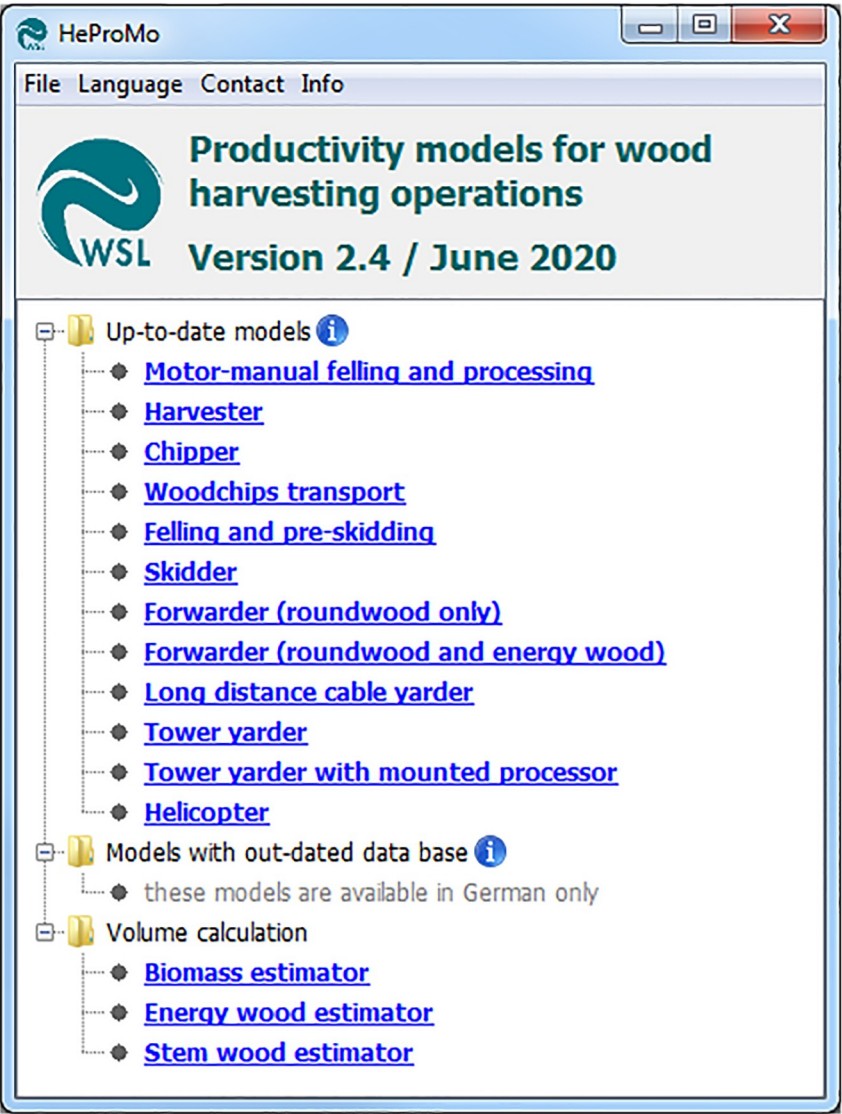

**Fig 4. Overview of HeProMo user interface.**

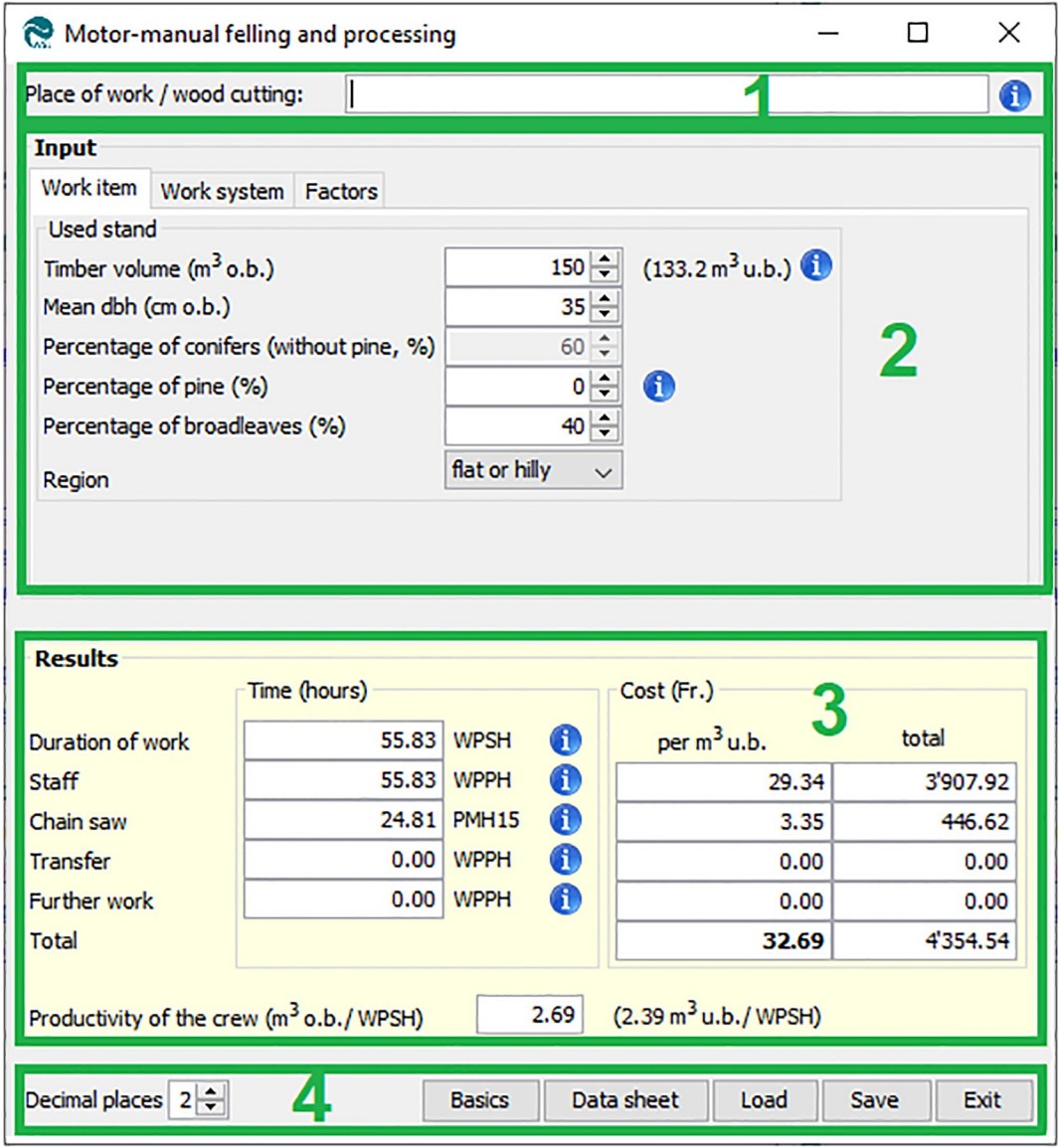

**Fig 5. GUI structure of FO models.**

**3.2.2 Forest operation models.** All models representing FOs have the same structure, which comprises four elements (Fig 5). The first element is a text field in which the current FO can be described. This step is optional but can be useful to identify a specific harvesting site if results are saved or printed out.

The second element is the input area. It consists of the tabs "work item", "work system", and "factors". The "work item" tab contains input parameter fields related to trees, timber, opening-up, and terrain. The most important input field in all models is the volume of wood to be processed. Other input fields are the mean diameter at breast height (dbh), share of conifers, and terrain slope. The second tab, "work system", consists of input fields for labor and machinery as well as for the related cost unit rates. In addition, input fields for the daily work time can be found there. The "factors" tab allows the change of additional model parameters,

such as the conversion factor from over to under bark or a forest-enterprise-specific correction factor to adapt the model to specific conditions. The currency code can be adapted here too.

The third element is the result section. For all models, it consists of three columns displaying time (cf. section 2.4.1), costs per cubic meter, and total costs for positions such as labor or machinery.

The forth element consists of the following items of control:

- Decimal places: allows to adapt the number of decimal places in the results section.

- Basics: opens the documentation of the current model. For most models, the documentation comprises two parts; part A, which describes the model itself (modeled harvesting process, data base, description of input variables, etc.), and part B, which describes and discusses statistics and modeling.

- Data sheet: creates a pdf-file that contains all input variables and the results of the current calculation.

- Save: saves the current input data to a hpm-file.

- Load: reloads input data from a previously saved hpm-file.

- Exit: closes the current model.

Throughout the application, blue info-buttons provide additional information where necessary. Hovering the mouse over such a button displays the additional information in a tooltip.

**3.2.3 Volume estimation models.** The three models for volume estimations (biomass, energy wood, and stem wood estimator; cf. section 2.1) have a layout similar to that of the FO models. However, they omit separate tabs for the input fields because they have fewer input and output variables (Fig 6).

## 3.3 API access

An API allows programmatic access to an application by code of another application. In the last few years, HeProMo was embedded into the code of several other applications, such as applications of the Swiss NFI. Before the API was available, this was done individually for each project with the help of the HeProMo developer. However, considering the peculiarities of each project, an individual integration is cumbersome to maintain and not sustainable. Therefore, there was a need to create a single well-documented API.

The design of the API focused on five goals: i) clear method names that disclose the precise unit expected or returned, ii) clear documentation of default values, iii) compatibility with programming languages other than Java, iv) performance (high throughput), and v) international usability.

The first goal, clear method names that disclose the precise unit expected or returned, is important because the units of some variables are not self-evident. This includes variables such as volume of wood (over or under bark), transportation distances (meters, kilometers, or miles), cost rates (e.g., of productive worktime or workplace time), and terrain slope (percentage, decimal, or category). This goal was achieved by suffixing each method name with the corresponding unit, e.g., "_ob" for over bark, "_m" for meters, or "_category" for categorical variables.

The second goal, a clear documentation of default values, is important because a default value is assumed for each variable that is not explicitly set. While this behavior is unproblematic in the GUI version because all input variables are directly visible to the user, an API needs a clear documentation of such values to avoid becoming a black-box. This goal was achieved

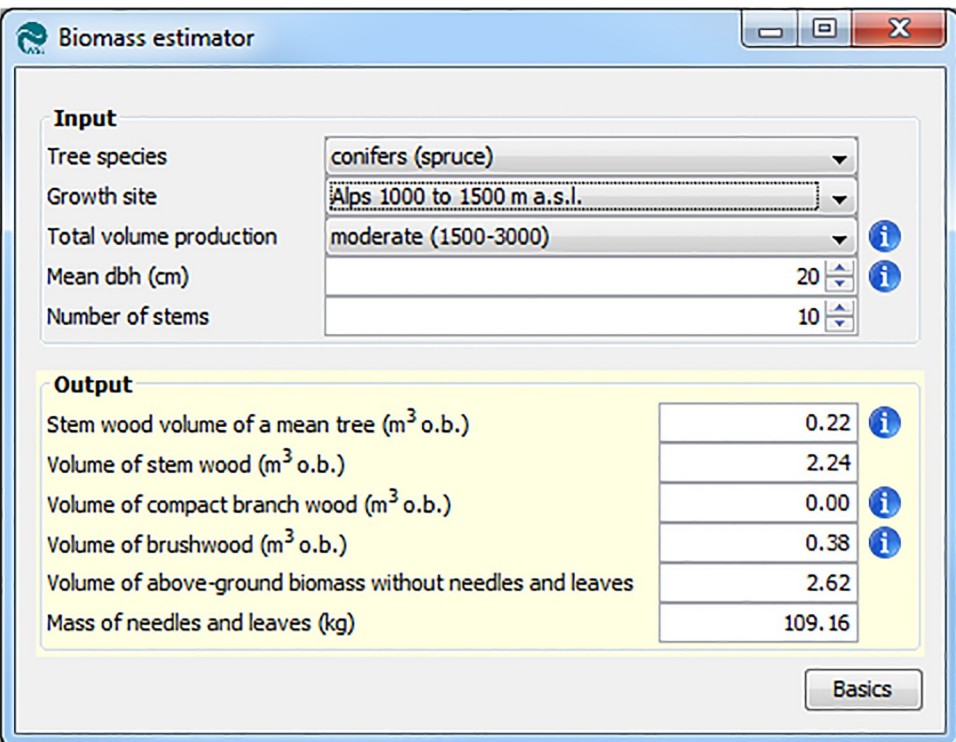

**Fig 6. Screenshot of the biomass estimator.**

by having a constant in the code defining the default value for each variable. The values of these constants are visible in the Javadoc (an API documentation in HTML format that can be created directly from the Java source code) and are currently the same as in the GUI version.

The third goal, compatibility with programming languages other than Java, is important because the projects that need to access the HeProMo code are written not only in Java but also in languages such as Python or SAS. These languages have a different type system than Java. Therefore, a mechanism to translate types between these languages and Java is needed. In HeProMo, this affects in particular methods with arguments of type *enum*, which is used for categorical variables. This goal was achieved by providing for each HeProMo model a dedicated Java class that acts as a wrapper class to the internal code. Such a class has first all the default values defined in constants, followed by all setter- and getter-methods. All methods in these classes use solely the primitive Java type *double* as method argument or return type. This simplifies the use of the class by other programming languages. Therefore, for example, if a method internally needs the terrain slope as an argument of type enum (i.e., the slope is a categorical variable), the API takes a value of type double, rounds it to the closest integer, and translates that integer into the corresponding internal enum value. The mapping of numbers to categories is described in the Javadoc of the method. Listing 1 shows an example of how the API can be used with Python.

The fourth goal, performance, is important because the API is designed to be usable for scientific purposes, where often a large number of calculations must be processed. This goal was achieved by using a lazy-evaluation approach that reduces the number of calculations to a minimum. Setting an input variable does not trigger a calculation, and getting a result variable only triggers a calculation if an input variable has changed since the last calculation. A single calculation provides *all* result variables, as most individual result variables depend on

productivity, and calculating the productivity is the computationally most expensive part of the entire calculation process.

The fifth goal, international usability, is important because HeProMo is currently used mainly in German-speaking countries of Central Europe, but the software should also be usable for English-speaking users. This leads to a trade-off regarding the language of the Java class, method, and variable identifiers in the API. All identifiers in the source code were in German so far. Providing only an English API for often highly domain-specific technical terms would be an unnecessary burden for German-speaking API users, as the English translation of well-known technical terms in German is often not straight-forward and can lead to confusion. Therefore, two versions of the API are provided. In addition to an API with German identifiers, we implemented a second version of the API with class, method, and variable identifiers in English for international API users. These classes, which are located in a separate package, were implemented by using the adapter pattern [26] and directly forward calls to the corresponding classes with the German identifiers.

```
# import py4j to connect to the HeProMo-API written in Java
from py4j.java_gateway import JavaGateway
# connect to the JVM
gateway = JavaGateway()
# get a reference to the Java class HeProMoEntryPoint
hepromoEntryPoint = gateway.jvm.ch.wsl.fps.hepromo4python.
HeProMoEntryPoint()
# get model for motor-manual felling
model = hepromoEntryPoint.getSimpleMotorManualFellingAndPro-
cessing2014()
# set some input variables
model.setTimberVolume_m3ob(300.0)
model.setMeanDbh_cm(30.0)
# print output
print(model.getProductivity_m3obPerWPSH())
```

**Listing 1: Use of the HeProMo-API with Python.** This example requires a small helper tool available at https://github.com/holmstefan/hepromo4python/releases/.

## 3.4 Software architecture and testing

The application is completely written in Java and uses Swing as the GUI toolkit. The source code is divided into four packages, namely hepromo.api, hepromo.gui, hepromo.model, and hepromo.util. The model package contains the core functionality and should not be directly accessed by external code as its interface may change—in contrary to the api package, which is designed to remain stable. The model package is completely independent of the gui and api packages. Therefore, the software could be distributed without the gui or without the api package if necessary. The classes in the model package follow closely the presentation of the models in the GUI, i.e., each model has a class for work item, work system, and factors. Further, each model has a calculator-class that transforms the input variables into a result-object containing all output variables. To create pdf files, the Apache FOP library [27] is used.

All models have been tested using TestNG [28] by applying a data-driven testing (DDT) approach. There is a csv-file for each HeProMo model containing the test cases for that model. This csv-file contains a test case on each row: the first columns contain input variables, and the later columns the expected output variables. With this approach, approximately 25'000 output values are verified whenever the test suite is run.

In addition, AssertJ Swing [29] is used to perform the GUI tests. The goal of these tests is to assure that the unit of an output value displayed in the GUI corresponds to the unit that was actually calculated for that value. This was introduced because the units (e.g., for the harvesting productivity) sometimes slightly differ across different models.

## 4. Discussion

### 4.1 Business use

For business use, such as a one-time estimation of time and costs of a FO for a single stand before realizing the FO, the use of the GUI version of HeProMo is recommended.

As all input fields have a carefully selected default value, a preliminary calculation result is already shown when the FO model is opened in the application. However, two kinds of input variables need to be adapted in most cases: those specific to the FO site and those specific to the company.

The most important input variable related to the FO site is the volume of wood to be harvested or processed. Additional input values such as mean dbh, share of conifers, and terrain slope increase the accuracy of the productivity estimation.

Values such as cost unit rates of labor and machinery are company-specific. Therefore, these values should be adapted and saved, so they can be loaded again the next time HeProMo is used. Another company-specific value is the *specific correction factor*, which can be set in each model. This factor allows the model to be adjusted for company-specific conditions, for example, if a company has a systematically lower or higher productivity for a certain FO due to slightly deviating harvesting methods or the use of different machinery.

Input fields should be self-explanatory. In case of possible ambiguities of input variables, there is an info button next to the input field. The equal input and output field structure in all models simplifies the use of the software. Finally, all models are accompanied by a manual, where all the basics and mathematics behind the model are documented in detail.

### 4.2 Scientific use

Increasing scientific applications of HeProMo are a result of the growing need to quantify the sustainability impacts of wood use. For this purpose, HeProMo is a suitable tool to estimate both monetary and environmental impacts of timber harvesting. For the latter, it can provide the basis to relate emissions and resource consumption of FOs to their performance. Therefore, HeProMo can be applied in life cycle assessments (e.g., [30]) and has been used to develop forestry standards that are used in the life cycle inventory database ECOINVENT [31]. The applications for scientific studies and in the context of demanding practical applications are mostly of a more complex nature and require the processing of bulk data.

**4.2.1 Example of bulk processing.** Contrary to business use, scientific use of such models often require the estimation of not only a single FO, but a large set of different FOs with varying input parameters. This is done, for example, in the context of the Swiss NFI [22, 32]. Within NFI, HeProMo is used to estimate timber harvesting costs for NFI sample plots [33], for the simulation of forest management scenarios [34, 35], and in further ongoing projects. For this kind of applications, a possibility to access the HeProMo models programmatically is essential.

However, it was difficult to integrate the earlier version of HeProMo programmed in.NET into the NFI applications written in SAS. Therefore, the core algorithms of HeProMo needed to be translated into SAS code [33]. With the new Java-HeProMo and its API, both projects written in Java and in other languages, such as SAS or Python, can now directly access the

**Table 3. Studies using HeProMo for study regions outside Switzerland.**

| Reference | Study region | HeProMo version used | HeProMo models used |
|---|---|---|---|
| [36] | German | Java version | Motor-manual, forwarder, skidder |
| [30] | Germany | Java version | Harvester, forwarder |
| [37] | Austria | .NET version | (Harvester, skidder, chipper, cable cranes) |
| [38] | European Union (EU28) | Directly used formulas from documentation | Motor-manual felling with chainsaw |
| [39] | France, Italy, Slovenia | .NET version and Excel version | Cable cranes (harvester, forwarder) |
| [20] | Germany | .NET version | Forwarder, skidder, chipper (motor-manual, harvester, cable cranes, helicopter) |
| [40] | Austria | .NET version | Motor-manual, cable cranes |

A model name in brackets indicates that the model is either not explicitly mentioned but assumedly used or that the model is mentioned but it is uncertain if it was actually used.

HeProMo code through its API. This emphasizes the value of the new HeProMo implementation, as the clearly documented API facilitates the programmatic use of HeProMo.

**4.2.2 Examples of international use.** A literature review revealed that published studies referring to HeProMo currently (2020) mainly cover study regions in Switzerland. However, a few studies from other countries and research institutes have been identified (Table 3). Five out of seven studies were conducted in German-speaking countries. Since 2016, HeProMo is available not only in German but also in English, French, and Italian, which is expected to facilitate its use in other countries.

## 4.3 Other software for estimating productivities and costs

Few software tools that allow for a direct estimate of wood harvesting productivities are described in the literature. Examples of such tools are the STHARVEST [41], IRLPACA [42], and HCR Estimator [43, 44]. However, none of them were developed for Central European conditions. Another category of tools (e.g., [45, 46]) use the k-nearest-neighbors-algorithm to estimate FO productivities [47]. This algorithm is especially suitable for forest enterprises and private forest owners to estimate productivities based on their own data records of former FOs.

In addition to such software tools, numerous scientific articles present formulas to calculate productivities and in most cases also costs for all kinds of FOs and under various conditions (e.g., [48–51]). A cross-section and comparison of 27 studies of this kind can be found in Hiesl and Benjamin [52]. However, these software tools and formulas are always specific to particular FOs and conditions. Mostly, they refer to FOs using CTL systems with harvesters and forwarders. HeProMo complements the existing tools by allowing productivity estimation of 12 kinds of FOs under Central European conditions. In particular, there are motor-manual processes included, which are still often used in the Central European Alps due to the difficult terrain.

## 4.4 Environmental relevance

In addition to economic aspects, performance indicators have been increasingly focusing on energetic inputs and greenhouse gas (GHG) emissions of FOs to enhance their environmental performance (e.g., [53–55]). The most recent and comprehensive reviews were provided by Klein et al. [56], Cosola et al. [57], and Đuka et al. [58]. Most emissions caused by FOs depend on the fuel consumption of machines (e.g., [59]), and fuel consumption strongly correlates with machine productivity (e.g., [54, 60, 61]). Therefore, HeProMo can be used to optimize processes from an environmental viewpoint once the FO productivity is known.

## 4.5 Limitations

There are some constraints in using the HeProMo productivity models. Due to the data base, each model is built for a specific harvesting operation. In practice, harvesting operations can occur under conditions (terrain, machinery, operational procedure) that do not correspond exactly to the conditions of the FOs used for modeling.

Often, forest enterprises and private forest operators try to perfect the harvesting operation, e.g., by using additional machinery. If so, the productivity model can be adapted to their needs by using a specific correction factor (see 4.1).

The HeProMo productivity models are based on data of FOs in Central European forests, mainly in Switzerland and Germany. There, FOs occur in forests treated by specific sylvicultural methods. The main characteristics are single tree harvesting, mixed and uneven-aged stands (conifer and broadleaves), and many environmental restrictions, e.g., soil protection on skid trails. Therefore, for other conditions such as clear-cuts and few environmental restrictions, the productivity model would not fit best.

## 4.6 Outlook

The current HeProMo is the result of significant developing efforts over the past two decades. To prevent the software from becoming outdated and the loss of invested work, continuous maintenance and improvement of the software are essential. Possible directions for the future of HeProMo are outlined in the following paragraphs.

In addition to the continuous general maintenance of the software, it is planned to continuously further develop HeProMo by adding additional models or adapting existing ones whenever necessary. The goal of going open-source is to support such enhancements and make the software even more transparent. It offers the opportunity for everyone to extend HeProMo, for example, by adding additional models, fine-tuning existing ones according to the user's needs, or translating the application into other languages. Finally, it mitigates the risk of the source code of this long-standing project getting lost over the years. The source code and further relevant files of HeProMo are stored on GitHub. Interested users have several possibilities to influence the further development of the tool, such as i) creating an "issue" to request a new feature or report a bug, ii) adapting the code and sending a "pull request" so we can integrate their code into ours, or iii) creating a "fork", i.e. copying the current code into a new project and continuing the further development at their own discretion.

As the basis of an empirical model is statistical data, collecting up-to-date data is an important prerequisite for the future development of HeProMo. Updating the models is often difficult because there are no data available from forest enterprises or forest operators. In the future, data collection processes could be improved by establishing a permanent monitoring system not yet installed and automated data-collecting methods.

Finally, in the future, HeProMo could be extended vertically and horizontally. Vertically, by integrating relevant ecological effects of FOs into the models (in addition to productivity, time, and costs). Horizontally, by integrating HeProMo into a library of models covering not only harvesting operations, but also operations related to planting, tending of young forest and forest stands, and forest protection.

## 5. Conclusions

This article described the development and application of HeProMo, an open-source software to estimate the wood harvesting productivities of 12 different kinds of forest operations under Central European conditions. Having a collection of models with a common structure and the possibility of a programmatic access via an API is an important prerequisite for the calculation

of productivities and costs of forest operations in scientific applications such as in a national forest inventory. While forest enterprises could rely on their experience to estimate their individual productivities, this is not an option for scientific applications, which need to calculate according to a common standard. Nevertheless, even for forest enterprises and practitioners, HeProMo offers new possibilities to enhance their processes by allowing estimations of forest operations before they are realized, with a tool that has a solid data base and is easy to use.

After HeProMo evolved and was proven useful in science and industry for more than 20 years, it is now being made open-source. By making the code available to the scientific community and interested practitioners, we lay the foundation for further continuous improvement of the software and its use in a broad range of applications in science and practice.

## Acknowledgments

The authors would like to thank all data suppliers (cf. Table 2), without whom it would not have been possible to create such a collection of forest operation models.

## Author Contributions

**Conceptualization:** Stefan Holm, Fritz Frutig, Renato Lemm, Oliver Thees.

**Data curation:** Fritz Frutig, Renato Lemm, Oliver Thees.

**Formal analysis:** Fritz Frutig, Renato Lemm, Oliver Thees.

**Funding acquisition:** Fritz Frutig, Renato Lemm, Oliver Thees, Janine Schweier.

**Investigation:** Fritz Frutig, Renato Lemm, Oliver Thees.

**Methodology:** Stefan Holm, Fritz Frutig, Renato Lemm, Oliver Thees.

**Project administration:** Fritz Frutig, Renato Lemm, Oliver Thees, Janine Schweier.

**Resources:** Fritz Frutig, Renato Lemm, Oliver Thees.

**Software:** Stefan Holm.

**Supervision:** Fritz Frutig, Renato Lemm, Oliver Thees, Janine Schweier.

**Validation:** Stefan Holm, Fritz Frutig, Renato Lemm, Oliver Thees.

**Visualization:** Stefan Holm, Fritz Frutig, Renato Lemm, Oliver Thees.

**Writing – original draft:** Stefan Holm, Fritz Frutig, Janine Schweier.

**Writing – review & editing:** Stefan Holm, Fritz Frutig, Renato Lemm, Oliver Thees, Janine Schweier.

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
