## [Decision Letter · Decision Letter 0]

21 Oct 2020

PONE-D-20-25452

HeProMo: A decision support tool to estimate wood harvesting productivities

PLOS ONE

Dear Dr. Holm,

Thank you for submitting your manuscript to PLOS ONE. After careful consideration, we feel that it has merit but does not fully meet PLOS ONE’s publication criteria as it currently stands. Therefore, we invite you to submit a revised version of the manuscript that addresses the points raised during the review process.

We look forward to receiving your revised manuscript.

Kind regards,

Marc Hanewinkel

Academic Editor

PLOS ONE

Journal Requirements:

Additional Editor Comments:

Both reviewers feel that the manuscript can be published after revisions. They provide detailed comments for that (please also see the commented files) - one reviewer suggests to add some information on the development history and background of the software.

Reviewers' comments:

Reviewer's Responses to Questions

**Comments to the Author**

1. Is the manuscript technically sound, and do the data support the conclusions?

Reviewer #1: Yes

Reviewer #2: Yes

2. Has the statistical analysis been performed appropriately and rigorously? 

Reviewer #1: N/A

Reviewer #2: N/A

3. Have the authors made all data underlying the findings in their manuscript fully available?

Reviewer #1: Yes

Reviewer #2: Yes

4. Is the manuscript presented in an intelligible fashion and written in standard English?

Reviewer #1: Yes

Reviewer #2: Yes

5. Review Comments to the Author

Reviewer #1: This is an interesting article and a piece of useful software for the community. HeProMo is a handy tool of rough decision and calculations in Forest Operations and useful for business and scientific applications. This article describes the approach, data basis and operation of a decision support system for the implementing software.

I recommend changing the Article type (type of contribution) in software description. The article is more a description of the basis, handling and function of the software HeProMo, which results from scientific work, than an independent scientific work.

The figures, tables and list are most clearly presented and correctly labelled. The title is adequate and informative. The literature is, for the most part, up-to-date and relevant.

Detailed and general remarks/questions:

(Table 1) Please specify ‘place at the forest road’ (skidder, Forwarder 2x). Did you mean truck-accessible road (Helicopter) or a forest road and you need an additional ‘Holz vorführen‘?

(Line 59) Please specify the costs of 100%. Wood costs at the roadside? Please add a source.

(Figure 1) The place of figure 1 seems to be wrong (maybe to line 145 after the first referencing)

(Figure 1) Explain figure 1: What is the used definition of data monitoring? What is the different to automatized time studies? What is automatization of data monitoring?

(Table 2) The place of table 2 seems to be wrong (maybe to line 137 after the first referencing)

(Table 2) If possible, add year of actualization and Influence factors of the models.

(Line 185) Is there a possibility to implement the actual conversation between CHF/USD/EUR?

(Line 316) For English, the API (variable identifiers, methods, class) was adopted. What about French and Italian?

(Line 422) What about the limitation of influence factors, e.g. individual operators performance, and exceptional situation (storm, bark beetle,…)? Should they take into account by the specific correction factor? Is something planned in the future?

(General) What about Versioning of the software? Are the (stable) APIs always follow the change of the main JAVA program immediately?

(General/line 434) How can the interested scientific community influence the further development of the program?

(Line 464) Should the community improve the software?

Reviewer #2: Dear Author/s,

The manuscript:

HeProMo: A decision support tool to estimate wood harvesting productivities

is an interesting work presenting development and application of new tool specifically dedicated to Central European forestry conditions. The manuscript is written in a good way, is understood, though some parts require rewriting (marked in the main file). Additionally, I would suggest to write more why this tool was developed, what was the reason of developing it and what actually the new tool replaces in practices (in fact hypothesis of job undertaken would be useful before the objective/s were formulated, conclusion should answer the objective/s more directly).

English corrections, prof-reading would improve the manuscript, too.

Detailed comments are in the main file to be considered for improvement of revised version.

Good luck with further work,

Reviewer

6. PLOS authors have the option to publish the peer review history of their article (what does this mean?). If published, this will include your full peer review and any attached files.

Reviewer #1: No

Reviewer #2: No

---

## [Author Response · Author response to Decision Letter 0]

2 Dec 2020

We would like to begin by thanking the editor and the reviewers for their time spent on evaluating this manuscript and for their many helpful comments. In the following segments, we explain how we have addressed each point raised by the editor and the reviewers. The statements of the editor and the reviewers are presented in bold font.

Journal Requirements:

We adapted the manuscript to meet the style requirements.

Additional Editor Comments:

Both reviewers feel that the manuscript can be published after revisions. They provide detailed comments for that (please also see the commented files) - one reviewer suggests to add some information on the development history and background of the software.

Our detailed answers can be found below.

Reviewer #1

This is an interesting article and a piece of useful software for the community. HeProMo is a handy tool of rough decision and calculations in Forest Operations and useful for business and scientific applications. This article describes the approach, data basis and operation of a decision support system for the implementing software.

Thank you for your very helpful comments.

I recommend changing the Article type (type of contribution) in software description. The article is more a description of the basis, handling and function of the software HeProMo, which results from scientific work, than an independent scientific work.

I removed the statement "Type of Contribution: Research article" in the paper itself, because it does not belong there. The type of contribution is an option that is selected during the submission process. PLOS ONE does not offer the possibility to declare a manuscript as a software description in the submission process. I agree that the article describes different aspects of the software. However, I think that the article is an independent scientific work, because it also describes the method how this software was developed in detail (e.g. data collection, modeling approach), compares the software with other existing software, and declares where the software fills a gap of knowledge.

The figures, tables and list are most clearly presented and correctly labelled. The title is adequate and informative. The literature is, for the most part, up-to-date and relevant.

Detailed and general remarks/questions:

(Table 1) Please specify ‘place at the forest road’ (skidder, Forwarder 2x). Did you mean truck-accessible road (Helicopter) or a forest road and you need an additional ‘Holz vorführen‘?

We meant truck-accessible road and clarified this in the manuscript.

(Line 59) Please specify the costs of 100%. Wood costs at the roadside? Please add a source.

We added a clarification in brackets.

(Figure 1) The place of figure 1 seems to be wrong (maybe to line 145 after the first referencing)

We moved the figure to the correct place.

(Figure 1) Explain figure 1: What is the used definition of data monitoring? What is the different to automatized time studies? What is automatization of data monitoring?

We adapted Figure 1 to avoid confusion (replaced "time studies" with "manual time studies", "data monitoring" with "permanent data monitoring", added an example, and removed the two "automation"-arrows). Manual time studies are conducted by observing the workers and machines during a forest operation with a stop watch. Permanent data monitoring is e.g. a nationwide survey where forest enterprises regularly report operational data.

(Table 2) The place of table 2 seems to be wrong (maybe to line 137 after the first referencing)

We moved the table to the correct place.

(Table 2) If possible, add year of actualization and Influence factors of the models.

We added the year of actualization to the table. We did not add the influence factors of the models, as this would make the table too complicated - many models have around 5 influence factors (plus around 10 common to all models), some of the older models many more (up to 27 influence factors).

 (Line 185) Is there a possibility to implement the actual conversation between CHF/USD/EUR?

At the moment no. The currency code in the software is just a character string that can be changed to an arbitrary value. As cost unit rates differ across countries and regions (even if the currency is the same), even if we would actually have the possibility for a conversion, the cost unit rates still need to be adapted to local conditions, making the conversion useless. We reformulated that sentence in the manuscript.

 (Line 316) For English, the API (variable identifiers, methods, class) was adopted. What about French and Italian?

We think that at the moment an API in English for international users should be sufficient. It is in our opinion unusual to have an API in multiple languages. The German API exists rather for historical reasons, because the German version of HeProMo was the one used most for years. However, if we get feedback from the community that an API in additional languages is a desired feature, we won't hesitate to implement that.

(Line 422) What about the limitation of influence factors, e.g. individual operators performance, and exceptional situation (storm, bark beetle,…)? Should they take into account by the specific correction factor? Is something planned in the future?

Differences in performance of individual operators are not considered, but it is assumed that all operators are sufficiently trained on the used machines. If the performance of the operators is systematically lower or higher than assumed in the models, then the specific correction factor can be used.

The models are currently not intended for exceptional situations, such as storm or bark beetle, because the underlying data was gathered in standard situations. Until now it was not planned to extend the tools in that direction, but if the demand for it exists, it would certainly be interesting to extend the tool in that direction.

(General) What about Versioning of the software? Are the (stable) APIs always follow the change of the main JAVA program immediately?

The software version described in the manuscript is version 2.4.3 (June 2020) and is the most up-to-date version. We added the version number to the software availability section.

The API always follows the changes of the software immediately. How it changes depends on the type of change:

- Changes in the calculation method (e.g. due to newer data from forest enterprises, added or removed influence factors, or a bug) -> Immediately reflected by the API. Such changes will be clearly documented and the older versions always remain available for backward compatibility.

- Changes of the name of an input variable (e.g. because of a spelling error or a wrong translation) -> a new identifier with the correct name will be added to the API, the old one will remain in the API for backward compatibility but marked as Deprecated (a Java annotation that tells the developer that a certain class or method shouldn't be used anymore).

So, in a nutshell, the API is designed to remain stable, but the calculation behind may change. If this is the case, it will be clearly indicated.

 (General/line 434) How can the interested scientific community influence the further development of the program?

The source code and relevant files of HeProMo are now stored on GitHub. There are multiple possibilities how the further development of the program can be influenced. For example, a "fork" is possible, which means that an interested user copies the complete source code to an own project, and continues further development at his own discretion. Or an interested user creates an "Issue", where he can request a new feature or file a bug. Or he adapts the code and sends a "pull request", which is a request to us to integrate code written by someone else into our code. We clarified this in a footnote.

(Line 464) Should the community improve the software?

We offer this possibility to the community by making the code open source. The community can improve the software (in addition to our efforts to do so), if desired.

Reviewer #2

Dear Author/s,

The manuscript:

HeProMo: A decision support tool to estimate wood harvesting productivities

is an interesting work presenting development and application of new tool specifically dedicated to Central European forestry conditions. The manuscript is written in a good way, is understood, though some parts require rewriting (marked in the main file). Additionally, I would suggest to write more why this tool was developed, what was the reason of developing it and what actually the new tool replaces in practices (in fact hypothesis of job undertaken would be useful before the objective/s were formulated, conclusion should answer the objective/s more directly).

English corrections, prof-reading would improve the manuscript, too.

Detailed comments are in the main file to be considered for improvement of revised version.

Good luck with further work,

Reviewer

Thank you for your very helpful comments. We considered most of your 31 comments marked in the main file, including a rewriting of the abstract and reorganizing table captions. However, we disagree with a few of your comments, and explain the reasons in the following:

[Comments in manuscript, lines 101/108] Replace are with were (2x), PLease use past tenses when describe methods, results, discussion and conclusion.

In our opinion, present tense is correct in the marked places because it describes the behavior of the software, i.e. it describes facts that are still true and continue to be true. For the rest of the manuscript, we checked the tenses again and believe they are correct, as we did not change them after language editing and they seem reasonable to us.

[Comment in manuscript, line 393] Please add why new tool was needed for Central European conditions? Where was the difference/gap that new tool was needed?

This is explained in the introduction and at the beginning of chapter two. In the introduction, we explain why such a tool is needed (lines 82-86 of the revised document with tracked changes) and where it fills a gap (lines 87-94 of the revised document with tracked changes). At the beginning of chapter two, we characterize the forests and the forest operations in Central Europe (e.g. single-tree thinning, small timber volumes per cut) to point out why forest operations in these regions are different than elsewhere (lines 105-110 of the revised document with tracked changes).

English corrections, prof-reading would improve the manuscript, too.

The manuscript was language-edited by a professional language editor before we submitted the manuscript to PLOS ONE.

[Comment in manuscript, Table 1] to be deleted. motor-manual means by chainsaw, this extra info can be deleted

While this information might be obvious to an expert in the field of forestry, parts of the paper might be more interesting to people with a computer science background, and we believe that this is a useful information for these people.

---

## [Editor Report · Decision Letter 1]

8 Dec 2020

HeProMo: A decision support tool to estimate wood harvesting productivities

PONE-D-20-25452R1

Dear Dr. Holm,

We’re pleased to inform you that your manuscript has been judged scientifically suitable for publication and will be formally accepted for publication once it meets all outstanding technical requirements.

Kind regards,

Marc Hanewinkel

Academic Editor

PLOS ONE

Additional Editor Comments (optional):

The suggested comments by the reviewers have been carefully taken into account. The manuscript can now be accepted.
---

## [Editor Report · Acceptance letter]

17 Dec 2020

PONE-D-20-25452R1 

HeProMo: A decision support tool to estimate wood harvesting productivities 

Dear Dr. Holm:

I'm pleased to inform you that your manuscript has been deemed suitable for publication in PLOS ONE. Congratulations! Your manuscript is now with our production department. 

Kind regards, 

on behalf of

Prof. Dr. Marc Hanewinkel 

Academic Editor

PLOS ONE